# Microcurrent and Gold Nanoparticles Combined with Hyaluronic Acid Accelerates Wound Healing

**DOI:** 10.3390/antiox11112257

**Published:** 2022-11-15

**Authors:** Carolini Mendes, Anand Thirupathi, Rubya Pereira Zaccaron, Maria Eduarda Anastácio Borges Corrêa, João V. S. Bittencourt, Laura de Roch Casagrande, Anadhelly C. S. de Lima, Lara L. de Oliveira, Thiago A. M. de Andrade, Yaodong Gu, Paulo Emílio Feuser, Ricardo A. Machado-de-Ávila, Paulo Cesar Lock Silveira

**Affiliations:** 1Faculty of Sports Science, Ningbo University, Ningbo 315211, China; 2Laboratory of Experimental Phisiopatology, Program of Postgraduate in Science of Health, Universidade do Extremo Sul Catarinense, Criciúma 88806-000, Brazil; 3Graduate Program of Biomedical Science, Herminio Ometto Foundation, Araras 13607-339, Brazil

**Keywords:** wound healing, iontophoresis, gold nanoparticles, hyaluronic acid, inflammation

## Abstract

This study aimed to investigate the effects of iontophoresis and hyaluronic acid (HA) combined with a gold nanoparticle (GNP) solution in an excisional wound model. Fifty Wistar rats (n = 10/group) were randomly assigned to the following groups: excisional wound (EW); EW + MC; EW + MC + HA; EW + MC + GNPs; and EW + MC + HA + GNPs. The animals were induced to a circular excision, and treatment started 24 h after injury with microcurrents (300 µA) containing gel with HA (0.9%) and/or GNPs (30 mg/L) in the electrodes (1 mL) for 7 days. The animals were euthanized 12 h after the last treatment application. The results demonstrate a reduction in the levels of pro-inflammatory cytokines (IFNϒ, IL-1β, TNFα, and IL-6) in the group in which the therapies were combined, and they show increased levels of anti-inflammatory cytokines (IL-4 and IL-10) and growth factors (FGF and TGF-β) in the EW + MC + HA and EW + MC + HA + GNPs groups. As for the levels of dichlorofluorescein (DCF) and nitrite, as well as oxidative damage (carbonyl and sulfhydryl), they decreased in the combined therapy group when compared to the control group. Regarding antioxidant defense, there was an increase in glutathione (GSH) and a decrease in superoxide dismutase (SOD) in the combined therapy group. A histological analysis showed reduced inflammatory infiltrate in the MC-treated groups and in the combination therapy group. There was an increase in the wound contraction rate in all treated groups when compared to the control group, proving that the proposed therapies are effective in the epithelial healing process. The results of this study demonstrate that the therapies in combination favor the tissue repair process more significantly than the therapies in isolation.

## 1. Introduction

Currently, the prevalence of non-healing wounds in developed countries is 1 to 2%, and this rate is significantly increased in underdeveloped countries. However, this problem affects the population in general and constitutes a public health problem due to high treatment rates and costs [1,2,3]. A wound is any injury that disrupts the structure of living tissue [4], and skin restoration involves the complex organization of diverse cell types, chemokines, and growth factors [5]. However, when this inflammatory response occurs in a slow and uncontrolled manner, as in the case of patients with vascular dysfunction, obesity, and diabetes, it leads to chronic inflammation, which can last for months or even years.

New materials and approaches are urgently desirable for wound healing since current treatments are closely restricted to bed hygiene and dressing changes; although essential, in some situations, these procedures are not able to promote adequate healing. Therefore, it is important to develop other approaches that accelerate the acute inflammatory and re-epithelialization process, thus avoiding complications and excessive expenditures in the health system [6,7].

A microcurrent (MC) or a microgalvanic current is an electrical current in the range of microamperes (μA), polarized and continuous, and it promotes a physical stimulus and is non-invasive, non-pharmacological, and low cost. According to Foulds and Barker (1983) [8], the transepithelial potential differential (TPD) of an undamaged (intact) epidermis is 10–60 mV. An unequal distribution of sodium ions (Na+) over the skin causes TPD [9]. A wound affects the intact skin epidermis and prevents TPD. MC has the ability to restore the electrical component of the tissue by allowing positive ions to flow from the undamaged skin to the wound margins, realigning the flow and aiding in tissue restoration [10].

Furthermore, through electrophoresis and electro-osmosis, MC allows the passage of charged and uncharged biomolecules across biological membranes. Iontophoresis is the collective name for these two processes. Ion channels and proteins that encourage cell contraction, as well as migration, orientation, and proliferation to the injured site, are only a few of the biological components with which MC interacts [11,12,13]. It is noteworthy that the endogenous electric fields generated by MC during wound healing play an essential role in fibroblast migration and collagen synthesis, increased local blood flow, and angiogenesis, which contribute to wound healing [14,15,16].

Currently, the development of nanomaterials, such as gold nanoparticles (GNPs), represents a promising approach to wound care [17]. Due to their focused delivery, lower toxicity, and higher absorption, GNPs have been frequently employed to transport compounds more effectively. GNPs are molecules that are simple to synthesize and functionalize, besides being biocompatible and having easy cellular permeability [18,19]. Additionally, they show promise because of their anti-inflammatory and antioxidant qualities, as well as because of how easily they can penetrate the barrier and open the stratum corneum. Several studies have shown that the topical application of GNPs increases the expression of fibroblasts and growth factors, such as VEGF and FGF, contributing to collagen formation, neovascularization, and granulation tissue formation and leading to faster wound closure [20,21].

Hyaluronic acid (HA) is a component of the extracellular matrix, and it has excellent benefits for the treatment of dermal and epidermal lesions [22]. Its capacity to promote the production of fibrin, phagocytic activity, neutrophil, and macrophage mobility and the release of chemotactic factors to fibroblasts, which aid in the granulation phase, collagen synthesis, and ultimately wound healing, makes it useful in clinical practice [23,24]. One of the main problems with treating epithelial lesions is preventing the healing process from taking too long, and studies have shown new methods for wound healing dressings, such as electrospinning methods, biomaterials based on conductive polymers, carbon nanomaterials or inorganic nanomaterials, and biotextiles. These methods demonstrate great potential in wound healing and tissue engineering, as well as effectively guiding migration and promoting fibroblast proliferation [25,26,27,28]. However, this study aims to evaluate whether the combination of an electrophysical agent (MC) potentiates the effects of HA combined with GNPs and, thus, accelerates the inflammatory process, anticipating tissue repair phases.

## 2. Materials and Methods

### 2.1. Animals

All experimental procedures involving animals were performed in accordance with the Guide for the Care and Use of Laboratory Animals of the National Institutes of Health (Bethesda, MD, USA) and with the approval of the Ethics Committee of the university (Universidade do Extremo Sul Catarinense—UNESC) with protocol number 16/2020. All animal experiments comply with the ARRIVE guidelines.

Fifty Wistar male rats (60 days old, weighing between 250 and 300 g) were used, and they were kept at a controlled room temperature between 20 ± 22 °C, under a light–dark cycle 12/12 h, and with free access to water and food. Male rats were used so that there were no changes in the inflammatory analysis due to hormones.

The animals were randomly assigned to five experimental groups with n = 10/group: excisional wound (EW) group—without local or systemic treatment; the EW + microcurrent treatment (EW + MC) group (300 µA, 20 min); the EW + MC + 0.9% hyaluronic acid (EW + MC + HA) group; the EW + MC + GNPs (EW + MC + GNPs) group (20 nm, 30 mg/L); and the EW + MC + HA + GNPs group.

### 2.2. Excisional Wound Model

The wound model was made using a circular excision as described by Mendes [21]. The animals were anesthetized with 4% isoflurane. The dorsal region of each animal was shaved, cleaned, and disinfected with 70% alcohol. The mid-dorsal region, between the interscapular line and the tail, was removed with a circular surgical incision of approximately 2 cm in diameter. The wounds were uniform in diameter, depth, and location. 

### 2.3. Treatment

The animals received microcurrent treatment using a Ibramed Striat Esthetic^®^ (Ibramed, São Paulo, Brazil) device with the function of a microgalvanic (µA) current lasting 20 min. Two 1.0 cm^2^ electrodes (a positive pole and a negative pole) were used, resulting in 300 μA dosimetry. The gel containing HA (0.9%) and/or GNPs (30 mg/L) was applied to the electrodes as a carrier of the electric current (2 mL).

The first treatment session started 24 h after the injury, and the others were performed daily until the seventh day. The area treated with iontophoresis was the dorsal region, which was submitted to injury and trichotomy, using electrodes positioned at the edge of the wound. The animals were anesthetized during treatment with the inhalation of isoflurane 4% to ensure immobilization and the correct application of treatments.

### 2.4. Euthanasia

The animals were euthanized 12 h after the last application of the MC, HA, and/or GNPs. After that, the external border of the wound was surgically removed and immediately processed and stored in a freezer at −70 °C for further analyses.

### 2.5. Synthesis and Characterization of the GNPs

Next, 20 nm GNPs were synthesized using the Turkevish method adapted by Della Vechia et al. [25]. The GNP solution was characterized using ultraviolet–visible (UV–VIS) spectroscopy with a spectrophotometry SpectraMax Plus 384 Microplate Reader (Molecular Devices, LLC; San Jose, CA, USA). The average hydrodynamic diameter and zeta potential of the GNP solution were determined using dynamic light scattering (DLS). For a TEM analysis, the GNP solution was placed on a carbon-coated copper grid (200 mesh) and dried for 24 h (room temperature) for further image acquisition. 

### 2.6. Wound Size Analysis

Digital images of the wounds were taken at a resolution of 3264 × 2448 pixels and analyzed using ImageJ^®^ 1.51 software (National Institute of Mental Health, Bethesda, MD, USA). For the visual verification of the evolution of the healing process and the measurement of the size (length and width) of the wounds, images were obtained on days 0 and 7 of the treatment, so the variation in the wound areas during this period in % could be calculated. 

### 2.7. Histomorphometry

The wound samples of four animals per group were conditioned for 48 h in a 10% formaldehyde-buffered solution (pH 7.4 phosphate buffer), followed by histological processing, and they were then embedded in paraffin. Thereafter, 5.0 µm sections were subjected simultaneously to respective hematoxylin and eosin (HE) staining in order to quantify the inflammatory infiltrate, blood vessels, and fibroblasts. Moreover, Ponceau S staining was performed to quantify the percentage area of total collagen compaction [29,30,31,32].

Furthermore, all histological sections were visualized using a LEICA^®^ DM-2000B optical microscope with a LEICA^®^ DFC-300 FX (Leica Microsystems, Wetzlar, Germany) camera connected to the computer with LAS^®^ software—Leica Application Suite (version 3.3.0)—in order to capture images.

For a simultaneous quantification (evaluated by a trained pathologist) of the inflammatory infiltrate and fibroblasts, the “Cell Counter” plugin in ImageJ software (5 images/animal/time/treatment in 400× magnification) was used, and for blood vessel quantification at 200× magnification, HE stain was performed. To quantify the percentage area of total collagen compaction (stained area in red: total collagen compaction stained by Ponceau S) at 400× (5 images/animal/time/treatment), the “Color deconvolution” plugin in ImageJ software was used (“Vector” <FastRed FastBlue DAB>) [33].

### 2.8. Determination of Cytokine Content Using ELISA

The samples were processed, and then the plate was sensitized for further incubation with the antibody. To measure cytokines (IFNϒ, TNF-α, IL-1β, IL-6, IL-4, IL-10, TGF-β, FGF), the enzyme-linked immunoabsorbent assay (Duoset ELISA) capture method (R&D system, inc., Minneapolis, MN, USA) was used.

### 2.9. Biochemical Analysis

Tissue determination of Reactive Oxygen Species (ROS) and nitric oxide: The production of hydroperoxides was determined by the intracellular formation of 2′,7′dichlorofluorescein (DCF) by ROS as described by Mendes et al. [21]. NO production was evaluated spectrophotometrically through the stable metabolite nitrite [21].

Tissue determination of oxidative damage marker levels: The oxidative damage to protein in the homogenized skin tissue samples was measured by the determination of carbonyl groups. The total thiol content was determined using the 5,5-dithiobis (2-nitrobenzoic acid) (2-nitrobenzoic acid) (DTNB) method [34].

Tissue determination of antioxidant defenses: SOD was measured in homogenized skin tissue by the inhibition of adrenaline oxidation, which was adapted from Bannister and Calabrese [35]. A standard curve constructed using reduced glutathione was used to calculate the GSH levels in the samples [36].

Protein Content: The protein content in homogenized skin tissue was measured following the protocol from Corrêa et al. [34].

### 2.10. Statistical Analysis

Data are expressed as the mean ± standard error, and they were analyzed statistically using one-way analysis of variance (ANOVA) tests, followed by Tukey’s post hoc tests. The significance level for the statistical tests is *p* < 0.05. Statistical Package for the Social Sciences (SPSS) (New Orchard RoadArmonk, New York, NY, USA).

## 3. Results

### 3.1. Physicochemical Properties of the GNP Solution 

To evaluate the formation and physical–chemical properties of the GNPs, UV–VIS, DLS, and zeta potential analyses were performed. The GNPs presented a maximum absorption peak at 524 nm and a ruby red color, indicating the formation of GNPs with a size of nearly 20 nm [29]. The DLS analyses confirmed an average hydrodynamic diameter near 20 nm (27 ± 5). The zeta potential analyses of the GNP solution demonstrated high stability with a negative surface charge (−38 ± 8) at pH 6.8. The TEM images (Figure 1) demonstrated that the GNPs had a near-spherical morphology and a size in a range from 20 to 30 nm. 

### 3.2. Analysis of the Wound Contraction Rate

We evaluated the wound contraction rate in %, represented by the reduction in the surface area (Figure 2), where all treated groups showed a significant difference compared to the control group.

### 3.3. Histological Analysis

In Figure 3 and Figure 4, representative images of the histological sections of the integumentary system can be seen. A decrease in the mean number of inflammatory infiltrates was found in the EW + MC group and the EW + MC + HA + GNPs group (Figure 3B). Regarding the mean number of fibroblasts (Figure 3C), an increase can be seen in the EW + MC and EW + MC + GNPs groups when compared to the EW group.

Despite analyzing the percentage of collagen area (Figure 4B), no significant difference was observed between the groups.

### 3.4. Evaluation of Pro-Inflammatory Cytokines

Figure 5 shows the protein levels of pro-inflammatory cytokines. In Figure 5A, INFϒ levels decreased in the EW + MC + HA group (*p* < 0.05) and in the combined therapy group (*p* < 0.01) when compared to the EW group. TNF-α levels (Figure 2B) also decreased in the EW + MC + HA group (*p* < 0.001) and in the EW + MC + GNPs and EW + MC + HA + GNPs groups (*p* < 0.0001). In Figure 5C, we can see that the groups treated with GNPs (EW + MC + GNPs and EW + MC+ HA + GNPs) had their IL-1β levels decreased (*p* < 0.05). Finally, only the group in which the therapies were combined had their IL6 levels decreased (Figure 5D) when compared to the EW group.

### 3.5. Evaluation of Anti-Inflammatory Cytokines and Growth Factors

The protein levels of anti-inflammatory cytokines are shown in Figure 6. The levels of IL-4 (Figure 6A) and IL10 (Figure 6B) were increased in the EW + MC + HA and EW + MC + HA + GNPs groups in relation to the EW group. 

The group treated with only the microcurrent (EW + MC) and the group in which the therapies were combined (EW + MC + HA + GNPs) showed a significant increase in the levels of TGF-β (Figure 6C) when compared to the control group (EW). Finally, FGF levels (Figure 6D) increased in the EW + MC + HA + GNPs group when compared to the EW group.

### 3.6. Intracellular Determination of Oxidants 

It can be seen that only the group treated with the combined therapies (EW + MC + HA + GNPs) had their oxidant levels of DCFH (Figure 7A) and nitrite (Figure 7B) decreased when compared to the control group (CE).

### 3.7. Markers of Oxidative Damage and Antioxidants

Carbonyl levels had a significant decrease in the EW + MC + HA + GNPs group compared to the EW group (Figure 8A). When the sulfhydryl content was evaluated (Figure 8B), we observed that the EW + MC + HA group, as well as the EW + MC + GNPs and EW + MC + HA + GNPs groups (*p* < 0.0001), had its levels increased (*p* < 0.001) when compared to the EW group.

Antioxidant defense was measured through reduced glutathione and superoxide dismutase levels. SOD levels (Figure 8C) showed a significant decrease in the combined therapy group (EW + MC + HA + GNPs). GSH levels (Figure 8D) had a significant increase in the EW + MC + HA + GNPs group when compared to the control group. 

## 4. Discussion

Wounds are a big problem in terms of social impact, and despite the wide range of therapies that can promote the repair of epithelial lesions, the incidence of wounds is progressively increasing, since the most susceptible populations, such as the elderly and people with diabetes, are growing [1], making wounds a major public health problem. The use of therapies such as HA combined with GNPs has already been studied by the current authors [21], demonstrating promising results for wound healing with the use of a laser as an electrophysical agent. In this study, we approached the combination of HA and GNPs as a therapy with the use of another electrophysical agent, microcurrents (MCs).

Clinical studies have reported positive wound healing results using electrical stimulation, and for a while, it has been utilized in clinical practice to speed up wound healing [37,38]. An MC is a specific type of microampere (μA) electrical stimulation that replicates currents generated in the body at the cellular level, and it is known to boost cell physiology and growth [10,39]. According to animal and human research, MCs may help minimize the duration of the inflammatory phase, boosting the pace of healing and decreasing edema [36]. The guided movement of cells inside an electric field is known as galvanotaxis. When induced by MCs, this movement is evident in leukocytes and macrophages, which are critical mediators in various phases of healing, as well as in keratinocytes, vascular endothelial cells, and fibroblasts, which are cells responsible for new tissue [40,41,42]. This results in decreased inflammation and increased blood flow, contributing to the repair process.

In line with what has been exposed, in the present study, a reduction in the levels of pro-inflammatory cytokines (IFNϒ, TNFα, IL-1β, and IL6) and inflammatory infiltrates was observed in the histological analysis, which demonstrates the ability of the combination of therapies to modulate and accelerate the inflammatory process, favoring the transition to the proliferative and remodeling phase.

Individually, each therapy promotes some mechanisms of action in the tissue, and, when combined, they can potentiate their isolated effects. MCs allow the transport of bioactive molecules when combined with some drugs in a process called iontophoresis [43]. Studies that compared MCs with other similar technologies in terms of transdermal drug permeation stated that iontophoresis guarantees the delivery of drugs in an optimized and controlled manner [40] through the mechanisms of electro-migration and electro-osmosis [44,45]. The effect induced by the currents causes a transient and reversible structural disturbance in he stratum corneum, increasing the permeability of the skin [46] by enlarging the pores in the skin barrier or inducing the creation of new pores [47]. Furthermore, the ability of MCs to regulate the levels of pro-inflammatory cytokines, such as IL-1β, and to attenuate the acute inflammatory response is explained by its ability to modulate COX-2 gene expression [48].

Likewise, the use of GNPs can act as an anti-inflammatory agent in the treatment of various models of tissue injury [49] by blocking the activation of nuclear factor kappa β (NF-κB) and combining with the Cys-179 component of I-kappa-β-kinase (IKK-β), thus decreasing the expression of pro-inflammatory cytokines, such as TNF-α, IL-1β, and IL-6 [50]. The capacity to physically engage with extracellular IL-1, neutralizing its binding to the membrane receptor and preventing the pro-inflammatory signaling cascade, is also a significant mechanism [51,52]. GNPs also have antibacterial properties, which trigger endogenous responses that are crucial for the promotion of wound healing, mainly through the modulation of the inflammatory phase and hemostasis [48]. Furthermore, recently, in the study conducted by Ni et al. (2019) [53], it was demonstrated that GNPs can control the inflammatory response through the regulation of macrophage phenotypes (M1 and M2) and, therefore, generate a microenvironment with reduced levels of inflammatory cytokines and a greater number of reparative cytokines.

Only in the group receiving combination treatment was it possible to see an increase in anti-inflammatory cytokines and growth factors, together with a decline in pro-inflammatory cytokines. According to Nguyen et al. (2017) [54], IL-10 inhibits the activity of macrophages with the M1 phenotype, which helps to reduce inflammation, lessen tissue damage, and promote healing. The macrophage phenotype shifts to a pro-repair M2 phenotype as the regeneration process advances, and cytokines, such as IL-10 and IL-4, are released, and growth factors, such as TGF-β, FGF, and EGF, are stimulated, playing an important role as part of the tissue recovery/repair mechanism, promoting the movement of fibroblasts and epithelial cells to the wound, thus initiating the proliferative phase [55]. In addition, changes in the electrodynamic field through electrophysical agents causing a change in the behavior of the electrical poles guide the healing process, stimulate the expression of VEGF messenger RNA (mRNA), and increase angiogenesis in endothelial cells. The release of these growth factors is important to promote the initiation of the proliferative phase [56].

The effects attributed to HA describe the modulation of pathways that include the suppression of pro-inflammatory cytokines and chemokines [57], in addition to improving membrane stability and favoring cell survival. HA interacts with its cell surface receptors, RHAMM; ICAM-1; and CD44, the main hyaluronan receptor, involved in the control of the exacerbated acute inflammatory response. According to certain studies, HA can speed up healing and activate particular responses in all cells engaged in the process, particularly in those engaged in fibroblast migration and proliferation and angiogenesis, through the signaling cascade induced by CD44 [21,58]. This increase in fibroblast proliferation is associated with the ability of HA to increase the expressions of growth factors, such as TGF and FGF [59], a result observed in this study when the therapies were combined.

By reducing the formation of reactive oxygen and nitrogen species, reducing oxidative damage, and stimulating the antioxidant system, the group that was subjected to the combination of the suggested treatments also succeeded in controlling the levels of oxidative stress. Senel et al. (1997) [60] state that antioxidants are crucial for the healing of ischemic skin wounds, whereas oxygen-free radicals are crucial for the postponement of the healing of ischemic wounds. Research has already suggested that MC treatment prevents mitochondrial dysfunction in models by enhancing respiratory chain activity, stabilizing mitochondrial membrane potential, and producing more ATP [43].

The capacity of HA to bind to the CD44 receptor, activate pathways involved in the control of cellular redox status, produce intracellular ROS, and chelate Fe^2+^ and Cu^2+^ ions—which are required for the Fenton reaction—can all be linked to its antioxidant activities. In the absence of these ions, highly reactive hydroxyl radicals with DNA cannot be generated [61]. In addition, studies have reported that HA treatment promotes the protection of granulation tissue from damage caused by oxygen-free radicals [62].

Likewise, as an antioxidant agent, GNPs exhibit high catalytic activity for free radical scavenging reactions, affecting the thiol bonds of keap1, causing a conformational change that allows the release of erythroid nuclear factor-related factor 2 (NRF2) to the subsequent transcription of cytoprotective genes, thus contributing to cellular homeostasis [63]. In order to trigger the protective response, GNPs can stimulate ROS-mediated redox signaling, including mitogen-activated protein kinase (MAPK) and nuclear factor kappa B (NF-B), and suppress reactive oxygen and nitrite species [64,65].

Since excessive levels of ROS in a wound pose a barrier to the healing process, our results show the potential of combined treatments (MC + HA + GNPs) to restore tissue redox homeostasis and to create a supportive environment for the growth of new tissue.

Our findings demonstrate that the groups who received MC in conjunction with the topical application of compounds had a reduction in the inflammatory infiltrate when compared to the lesion group, which was consistent with the decrease in pro-inflammatory indicators, a result that can be explained by the interactions of MC with a variety of cellular components, which allows the cells to perform contraction, migration, orientation, and proliferation [66]. This is a result of negative polarity, which accelerates the healing process by increasing epithelial cell movement. A cathode can be used as an active electrode to draw fibroblasts to the wound site and to increase collagen production.

In line with that previously discussed, when evaluating the wound contraction rate in %, this study shows that all therapeutic approaches can promote the healing process by causing the wound to shrink since stimulation by microcurrents promotes the proliferation of fibroblasts and myofibroblasts, increases collagen synthesis, and improves the organization of collagen fibers at the injury site. Furthermore, the application of electric fields and currents similar to those produced by the body has been demonstrated to significantly alter cell shape and function, promoting cell division and protein production [67].

Taken together, these results are in agreement with those found in the study conducted by Mendes et al. (2020), where the benefits of using HA in combination with GNPs were also demonstrated and where the use of electrophysical therapies, such as lasers and MCs, to potentiate the effects of these molecules was shown, contributing to tissue repair.

## 5. Conclusions

The proposed therapies, when combined, seem to enhance their mechanisms of action, with promising effects in the control of the acute inflammatory response, accelerating the transition to the proliferative phase and increasing wound contraction.

As a proposal for further studies, it is believed that, with longer analysis periods (between 14 and 21 days), the response of the combined therapy group (EW + MC + HA + GNPs) would have greater repercussions in the collagen area and, consequently, in reducing the size of the wounds.

## Figures and Tables

**Figure 1 antioxidants-11-02257-f001:**
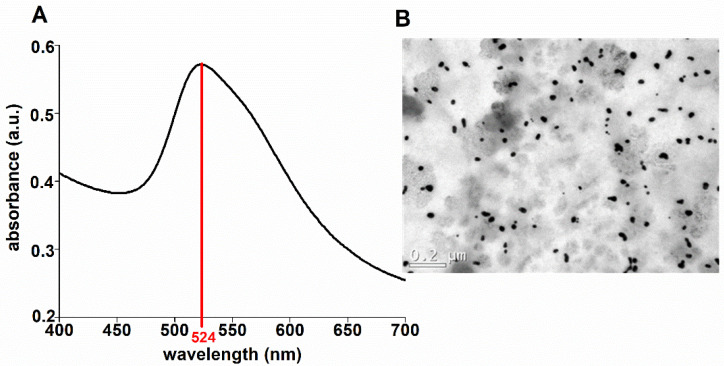
UV–VIS spectroscopy (**A**) and TEM images (**B**) of the GNP solution were obtained.

**Figure 2 antioxidants-11-02257-f002:**
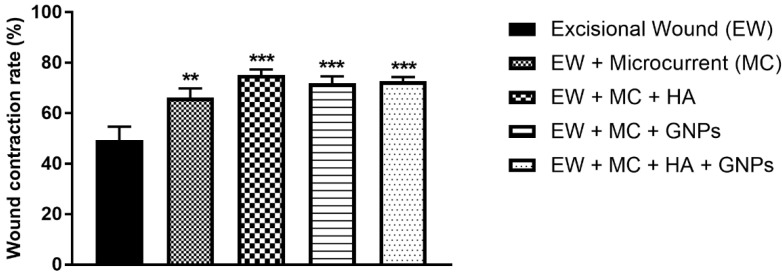
Effects of treatment with MC, GNPs, and hyaluronic acid on wound contraction. The data are presented in mean ± EPM, in which ** *p* < 0.01 vs. excisional wound group (EW) and *** *p* < 0.001 vs. excisional wound group (EW); (one-way ANOVA followed by Tukey’s post hoc test).

**Figure 3 antioxidants-11-02257-f003:**
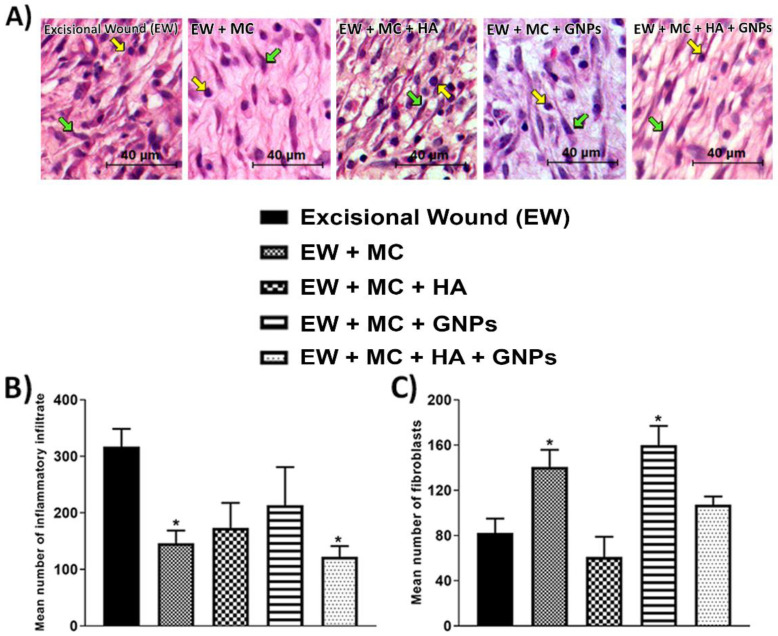
(**A**) Representative images of the histological sections of the integumentary system. (**B**,**C**) Effects of treatment with MC, GNPs, and hyaluronic acid on the number of inflammatory infiltrates (**B**) and fibroblasts (**C**). The data are presented in mean ± EPM, in which * *p* < 0.05 vs. excisional wound group (EW; one-way ANOVA followed by Tukey’s post hoc test).

**Figure 4 antioxidants-11-02257-f004:**
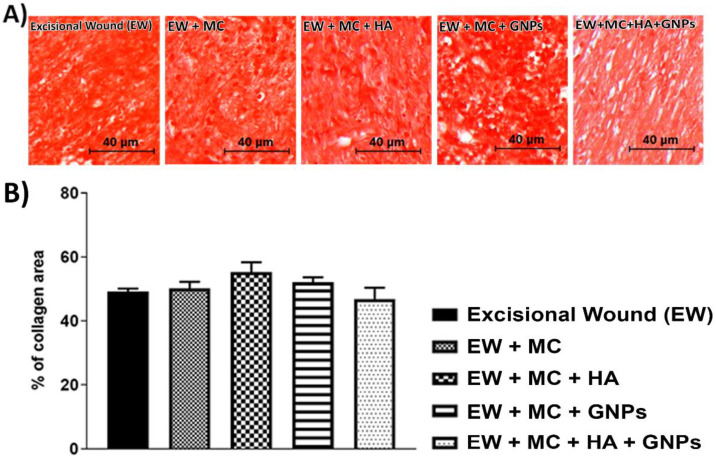
(**A**) Representative images of the histological sections of the integumentary system. (**B**) Effects of treatment with MC, GNPs, and hyaluronic acid on the % of collagen area. The data are presented in mean ± EPM, (EW; one-way ANOVA followed by Tukey’s post hoc test).

**Figure 5 antioxidants-11-02257-f005:**
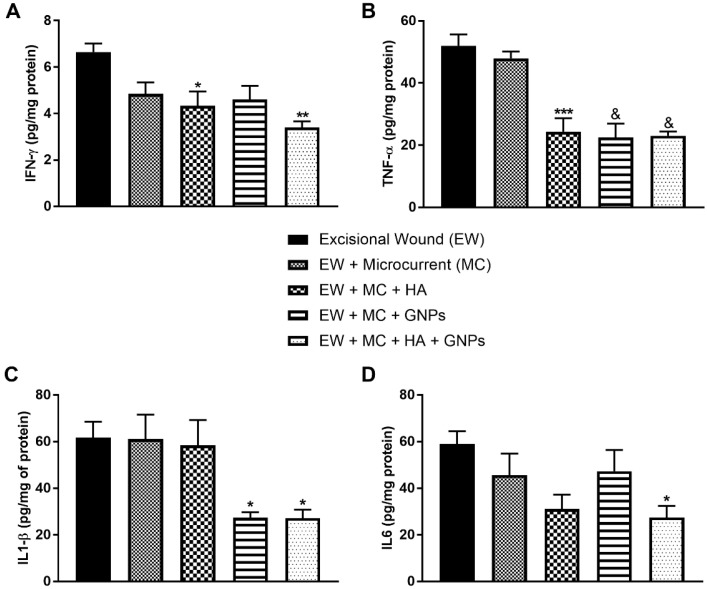
Effects of treatment with MC, GNPs, and hyaluronic acid on the protein levels of pro-inflammatory cytokines: (**A**) INF-ϒ, (**B**) TNF-α, (**C**) IL1-β, (**D**) IL6. The data are presented in mean ± EPM, in which * *p* < 0.05 vs. excisional wound group (EW); ** *p* < 0.01 vs. excisional wound group (EW); *** *p* < 0.001 vs. excisional wound group (EW); and & *p* < 0.0001 vs. excisional wound group (EW) (one-way ANOVA followed by Tukey’s post hoc test).

**Figure 6 antioxidants-11-02257-f006:**
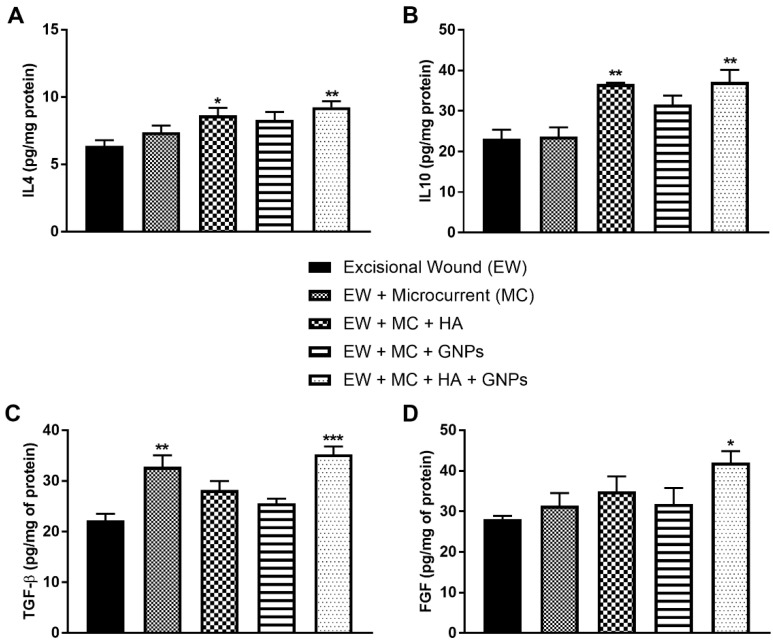
Effects of treatment with MC, GNPs, and hyaluronic acid on the protein levels of anti-inflammatory cytokines: (**A**) IL4; (**B**) IL10; (**C**) TGF-β; (**D**) FGF. The data are presented in mean ± EPM, in which * *p* < 0.05 vs. excisional wound group (EW); ** *p* < 0.01 vs. excisional wound group (EW); *** *p* < 0.001 vs. excisional wound group (EW) (one-way ANOVA followed by Tukey’s post hoc test).

**Figure 7 antioxidants-11-02257-f007:**
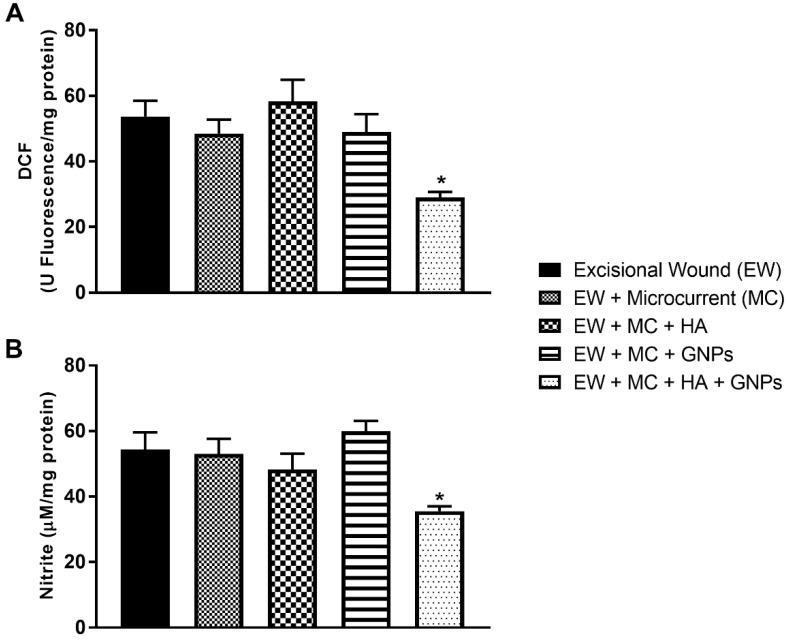
Effects of treatment with MC, GNPs, and hyaluronic acid on the levels of oxidants: (**A**) DCF; (**B**) nitrate. The data are presented in mean ± EPM, in which * *p* < 0.05 vs. excisional wound group (EW) (one-way ANOVA followed by Tukey’s post hoc test).

**Figure 8 antioxidants-11-02257-f008:**
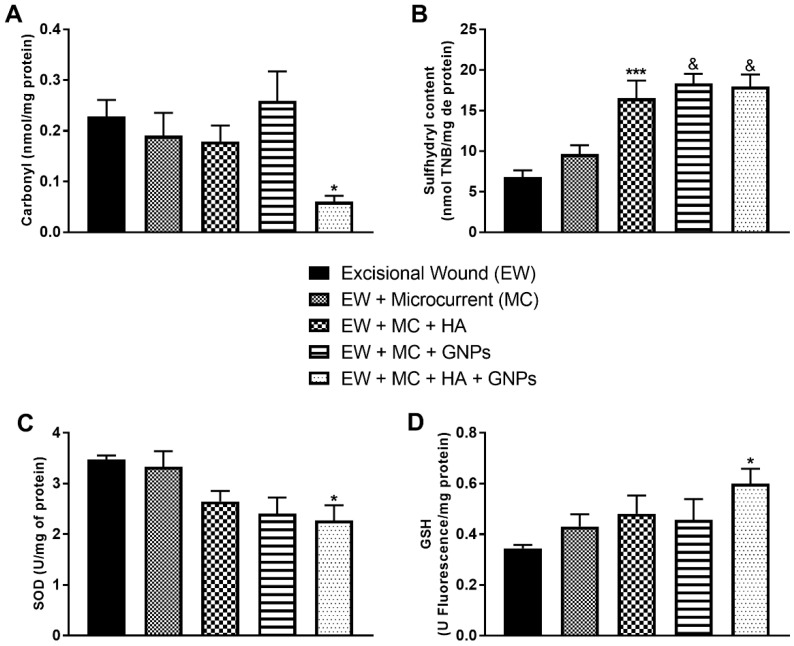
Effects of treatment with MIC, GNPs, and hyaluronic acid on the levels of oxidative damage markers and antioxidants: (**A**) carbonyl; (**B**) sulfhydryl; (**C**) SOD; (**D**) GSH. The data are presented in mean ± EPM, in which: * *p* < 0.05 vs. Excisional wound group (EW); *** *p* < 0.001 vs. Excisional wound group (EW); & *p* < 0.0001 vs Excisional wound group (EW); (one-way ANOVA followed by Tukey’s post hoc test).

## Data Availability

The datasets generated and/or analyzed during the current study are available from the corresponding author on reasonable request.

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
