# Peer review of "Microcurrent and Gold Nanoparticles Combined with Hyaluronic Acid Accelerates Wound Healing"

_antioxidants, 2022, doi:10.3390/antiox11112257_

Round 1

Reviewer 1 Report

The authors present an interesting paper about the use of gold nanoparticles on wound healing. Overall, the paper is interesting but the wound model needs clarification. The authors call it an "epithelial lesion" model but it seems as though they are creating a full-thickness wound (circular excision). They talk about "contraction" as a measure of healing, which supports more than an "epithelial" lesion. The authors should describe the wounding model more clearly and if it is full-thickness, then call it a full-thickness wound. For an epithelial lesion, one really studies re-epithelialization, not contraction. 

The data shows that there may be a decrease in oxidants but there really is no change in healing with any of the treatments. Therefore, the authors should not claim that the GNP process helps with healing. 

Finally, there are many abbreviations with their explanation in the abstract. Maybe that is OK with the journal but typically, the entire words are written out the first time they are used. 

Author Response

Reviewer 1

The authors present an interesting paper about the use of gold nanoparticles on wound healing. Overall, the paper is interesting but the wound model needs clarification. The authors call it an "epithelial lesion" model but it seems as though they are creating a full-thickness wound (circular excision). They talk about "contraction" as a measure of healing, which supports more than an "epithelial" lesion. The authors should describe the wounding model more clearly and if it is full-thickness, then call it a full-thickness wound. For an epithelial lesion, one really studies re-epithelialization, not contraction.

A: We appreciate the reviewer's suggestions to change the injury classification term, we have modified the way we refer to wounds throughout the text, legends, and figures, updating it to “excisional wound”. Regarding the wounding model, it was also tidied up and described more clearly as requested. All changes made are in yellow.

The data shows that there may be a decrease in oxidants but there really is no change in healing with any of the treatments. Therefore, the authors should not claim that the GNP process helps with healing.

A: Thank you for your comment. Yes, in fact, when observing some results of the group treated only with GNPs, as in the case of oxidants, we did not observe significant differences, but this difference was observed when the wound contraction rate (important data of the study) was evaluated, in the increase of the number of fibroblasts and some evaluated cytokines. However, it is evident that therapies when associated have better results, as together they were able to positively modulate these markers, as presented in the study discussion.

Finally, there are many abbreviations with their explanation in the abstract. Maybe that is OK with the journal but typically, the entire words are written out the first time they are used.

A: Thank you for your comment. The names of the abbreviations have been included in the abstract.

Reviewer 2 Report

Mendes et al. develop a new strategy through a combination of microcurrent, gold nanoparticles and hyaluronic acid for wound healing application. This manuscript was well written. Some minor revision should be conducted before publication.

1. The title is ambiguous and too long, and it should be rewritten.

2. Abstract must present the research results and contribution in a better and clear way. For instance, some important data should be presented in this section.

3. Some more descriptions should be added to introduce the recent advances of wound dressings in Introduction section, and further highlight the novelty of present study. Some important references like 10.1016/j.copbio.2021.08.019, 10.1016/j.apmt.2022.101542, 10.1007/s40820-021-00751-y, 10.1016/j.apmt.2022.101473, 10.3390/nano12050784 are missing, which should be discussed in this section.

4. The Y-axis should be revised as “Wound contraction rate (%)” to take place of “Wound contraction area” in Figure 2.

5. Figure 1 should be redrawn. The multiple components could not be seen in a clear manner.

6. Does it have any significant difference in Figure 4B?

7. The grammar and writing should be improved in the whole manuscript.

Author Response

Reviewer 2

Mendes et al. develop a new strategy through a combination of microcurrent, gold nanoparticles, and hyaluronic acid for wound healing application. This manuscript was well written. Some minor revision should be conducted before publication.

  1. The title is ambiguous and too long, and it should be rewritten.

A: Thank you for your comment. The title has been corrected as suggested by the reviewer: “Microcurrent and Gold Nanoparticles Associated with Hyaluronic Acid Accelerates Wound Healing”.

  1. Abstract must present the research results and contribution in a better and clear way. For instance, some important data should be presented in this section.

A: Thank you for your comment, we have addressed more results in the abstract, as requested by the reviewer.

  1. Some more descriptions should be added to introduce the recent advances in wound dressings in Introduction section, and further highlight the novelty of present study. Some important references like 10.1016/j.copbio.2021.08.019, 10.1016/j.apmt.2022.101542, 10.1007/s40820-021-00751-y, 10.1016/j.apmt.2022.101473, 10.3390/nano12050784 are missing, which should be discussed in this section.

A: As suggested by the reviewer, we added these references to the introduction, highlighting what is new for wound healing, and citing the importance of our study for the insertion of new treatments in this area.

  1. The Y-axis should be revised as “Wound contraction rate (%)” to take place of “Wound contraction area” in Figure 2.

A: Thank you for your comment, we took it under consideration and concluded that demonstrating the results in % is didactically better. We changed the text, captions, and figures.

  1. Figure 1 should be redrawn. The multiple components could not be seen in a clear manner.

A: Thank you for your comment. Microscopy image resolution has been improved. From the TEM it was only possible to verify the size and morphology of the GNPs associated with hyaluronic acid. It was possible to detect a denser cloud in the microscopy image, which may or not be related to the formation of a gel from hyaluronic acid.

  1. Does it have any significant difference in Figure 4B?

A: We have revised this and in figure 4B there’s no significant difference between groups.

  1. The grammar and writing should be improved in the whole manuscript.

A: Thank you for your comment, we have improved the grammar and writing in the manuscript.

Round 2

Reviewer 1 Report

Accept